# A homogeneous split-luciferase assay for rapid and sensitive detection of anti-SARS CoV-2 antibodies

Zhong Yao[1,2], Luka Drecun[1,3], Farzaneh Aboualizadeh[1,2], Sun Jin Kim[4], Zhijie Li [3], Heidi Wood[5], Emelissa J. Valcourt[5], Kathy Manguiat[5], Simon Plenderleith[6], Lily Yip [6], Xinliu Li[7], Zoe Zhong[7], Feng Yun Yue[8], Tatiana Closas[9], Jamie Snider[1,2], Jelena Tomic[1,2], Steven J. Drews[10,11], Michael A. Drebot[5], Allison McGeer[7], Mario Ostrowski[8], Samira Mubareka[6,8], James M. Rini[2,3], Shawn Owen [4,12,13 ✉] & Igor Stagljar [1,2,3,14 ✉]

Better diagnostic tools are needed to combat the ongoing COVID-19 pandemic. Here, to meet this urgent demand, we report a homogeneous immunoassay to detect IgG antibodies against SARS-CoV-2. This serological assay, called SATiN, is based on a tri-part Nanoluciferase (tNLuc) approach, in which the spike protein of SARS-CoV-2 and protein G, fused respectively to two different tNLuc tags, are used as antibody probes. Target engagement of the probes allows reconstitution of a functional luciferase in the presence of the third tNLuc component. The assay is performed directly in the liquid phase of patient sera and enables rapid, quantitative and low-cost detection. We show that SATiN has a similar sensitivity to ELISA, and its readouts are consistent with various neutralizing antibody assays. This proof-of-principle study suggests potential applications in diagnostics, as well as disease and vaccination management.

[1] Donnelly Centre, University of Toronto, Toronto, ON, Canada. [2] Department of Biochemistry, University of Toronto, Toronto, ON, Canada. [3] Department of Molecular Genetics, University of Toronto, Toronto, ON, Canada. [4] Department of Pharmaceutics and Pharmaceutical Chemistry, University of Utah, Salt Lake City, UT, USA. [5] Zoonotic Diseases and Special Pathogens Division, National Microbiology Laboratory, Public Health Agency of Canada, Winnipeg, MB, Canada. [6] Biological Sciences, Sunnybrook Research Institute, Toronto, ON, Canada. [7] Department of Microbiology, Mount Sinai Hospital, Toronto, ON, Canada. [8] Department of Laboratory Medicine and Pathobiology, University of Toronto, Toronto, ON, Canada. [9] Canadian Blood Services, Vancouver, BC, Canada. [10] Canadian Blood Services, Edmonton, AB, Canada. [11] Department of Laboratory Medicine & Pathology, University of Alberta, Edmonton, AB, Canada. [12] Department of Biomedical Engineering, University of Utah, Salt Lake City, UT, USA. [13] Department of Medicinal Chemistry, University of Utah, Salt Lake City, UT, USA. [14] Mediterranean Institute for Life Sciences, Split, Croatia. ✉email: shawn.owen@hsc.utah.edu; igor.stagljar@utoronto.ca

Laboratory testing is crucial for combating the coronavirus disease 2019 (COVID-19) pandemic[1,2]. Serological tests are used to determine the level of antibodies against severe acute respiratory syndrome coronavirus 2 (SARS-CoV-2) in blood[3]. Their results reflect the disease progression or its history, as well as the immunity of a patient[4], which is valuable information for diagnosis and disease management. Importantly, with the advent of different vaccines, serological testing is becoming a necessary tool for evaluating acquired immunity at both individual and population levels. In addition, serological testing is essential for epidemic studies and related policymaking.

Numerous serological assays have been developed[5]. Among them, lateral flow immunoassays (LFIAs) are rapid and easy to perform, and therefore have found use as point-of-care tests[6]. However, their lack of quantifiability, coupled with their relatively low sensitivity and specificity, limits their use as standard and reliable tests to evaluate antibody titers[7–9]. In contrast, enzyme-linked immunosorbent assays (ELISAs) are quantitative serological methods displaying good sensitivity and specificity[10,11]. They too, however, have notable shortcomings including long processing times (3–5 h), tedious procedures (multiple wash-aspirate cycles), and often extra steps to pre-process the binding plates. Several chemiluminescent immunoassay (CLIA) platforms targeting COVID-19 have also been developed by companies such as Abbott[12–14], DiaSorin[13,14], Roche[14], and Siemens[14]. These are highly automated assays suitable for centralized measurement of large content samples and are characterized by good quantifiability and sensitivity[14]. However, the measurements require highly specialized and expensive instruments, which limit their widespread application.

Here, we describe a liquid-phase serological assay for IgG antibodies against SARS-CoV-2 based on split tripart Nanoluciferase (tNLuc) that can be performed directly with patient sera. This assay, which we have called SATiN (Serological Assay based on split Tripart Nanoluciferase), displays quantifiability and sensitivity comparable to ELISA. It is also rapid/easy to perform and cost-effective, and it produces readouts consistent with neutralizing antibody tests. Taken together, these attributes strongly support its potential value in COVID-19 diagnosis as well as disease and vaccination management.

## Results

**Design of the SATiN assay.** Protein complementation assays (PCAs) are widely used to detect protein–protein interactions (PPIs)[15–19]. In these approaches, a "sensor" protein is split into two fragments, which are then fused to two candidate interacting proteins of interest. The binding of the two proteins of interest arranges the sensor fragments in a favorable position that allows them to reconstitute a functional protein which can produce a detectable signal representative of the PPI[20]. Different sensors such as fluorescent proteins, transcription factors, proteases, and more have been successfully used in various designs. Among them, split luciferases have been shown to have the advantages of high signal/noise ratio and rapid reconstitution, making them commonly used[21–23]. However, the conventional strategy of splitting luciferase into two fragments has limitations. For instance, the relatively large size of the fragments may interfere with target protein folding or function and/or the interaction with partner molecules. The residual intrinsic affinity between the two luciferase fragments may also lead to an increased background signal. A recently developed tri-part strategy circumvents these limitations by splitting NanoLuc® (NLuc), the brightest luciferase identified so far, into three fragments: two short peptides (β9 and β10 each containing 11 amino acids) and one 16 kDa fragment (Δ11S)[24,25]. Here we report the adaptation of this variant tri-part

NanoLuc® (tNLuc) for use in our SATiN COVID-19 antibody detection system (Fig. 1a). In our design, the β9 and β10 tags are separately fused to a pair of probes which can respectively recognize an IgG molecule against SARS-CoV-2 at different sites. The first probe is generated by fusing the β9 tag to the C2 domain of protein G, which exclusively binds to all the isotypes of human IgG but not to IgM, IgA, or IgE immunoglobulins[26]. The second probe is specific to antibodies that bind the SARS-CoV-2 spike (S) protein, the viral membrane protein responsible for host cell receptor binding[27] and which is also the target of most of the neutralizing antibodies found in patients[28]. We generated two forms of this probe by fusing the β10 tag to either the ectodomain of the S protein or to its receptor binding domain (RBD). The assay itself is straightforward to perform and involves only two simple steps (Fig. 1a): (i) diluted serum or plasma samples are mixed with the two probes and incubated for 30 min; (ii) an aliquot of this mixture is combined with the third component (Δ11S) and the luciferase substrate, and after 30 min luminescence is read with a luminometer. The whole process occurs directly in the liquid phase and does not require any washing steps.

As the peptide tags (β9 and β10) can be fused either at the N- or C-terminus of the probes or at both positions (Supplementary Fig. 1), we first sought to determine which combination of the various probes would result in the most signal with the least background. We tested all combinations of the probes with CR3022, an antibody that binds to the RBD of the SARS-CoV-2 S protein[29], and observed that all combinations produced signal, albeit to varying extents (Fig. 1b). This suggests that despite the different spatial relationships between the probes upon target antibody engagement, the molecules are sufficiently flexible to allow proximal localization of the two tags, enabling reconstitution of tNLuc into an active luciferase. Of all the combinations, two produced the highest signal-to-background: β9-G with β10-S and β9-G with β10-S-β10. Further testing of these two combinations using different concentrations of CR3022 demonstrated typical dose response curves (Fig. 1c, d). However, based on the calculated $K_d$ values using β10-S (0.42 μg/mL) and β10-S-β10 (0.23 μg/mL), we predicted that β10-S-β10 would have greater potential to detect antibody at lower concentrations (Fig. 1e). Thus, the β9-G/β10-S-β10 probe pair was chosen to be used in the final assay.

**Alleviation of the inhibitory effect of general IgG in SATiN.** Human plasma contains a high concentration of IgG (4–22 mg/mL, median 11 mg/mL in serum)[30,31], the vast majority of which will not be specific for the SARS-CoV-2 S-protein. Since they can, however, bind the protein G probe, these "non-specific" antibodies will reduce the sensitivity of our assay. We investigated the effect of IgG interference on the assay by adding different amounts of human IgG into the reaction mixture and as predicted inhibition was indeed observed (Fig. 2a). The detailed kinetics of inhibition were further analyzed based on the dose response testing of CR3022 in the presence of different amounts of IgG (Fig. 2b). We performed mathematical analysis by fitting different inhibition models to these experimental data. A model of allosteric noncompetitive inhibition provided the best fit, as judged by visual comparison of the curve positions/shapes with the plotted data points (Fig. 2b) and the calculated $R^2$ (0.9911). The allosteric effect might be derived from the binding of protein G to two sites on an IgG molecule while the detailed molecular mechanism of noncompetitive inhibition merits further exploration.

Binding parameters derived from the model ($K_i = 58.5$ μg/mL, and $K_d = 0.22$ μg/mL which is consistent with the results in

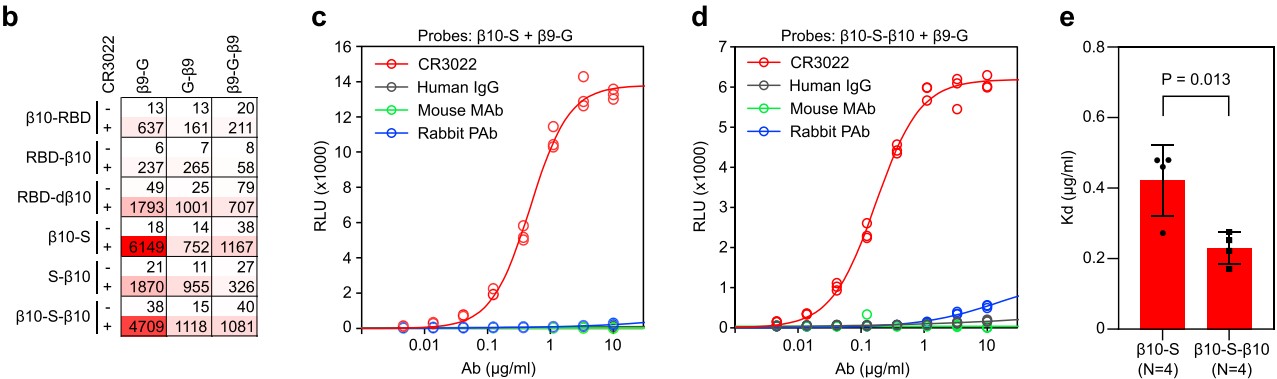

**Fig. 1 tNLuc-based SATiN assay for detecting α-SARS-CoV-2 antibody. a** Schematic workflow of the SATiN assay. **b** Scan of all probe formats/combinations in the SATiN assay using CR3022 Ab (2 μg/mL). Results are presented as a heatmap showing RLU values. Data shown here are a representative result of three independent experiments with similar results. **c**, **d** CR3022 (red) at different concentrations was tested with β9-G together with β10-S (**c**) or β10-S-β10 (**d**) probes. Human IgG (gray), a mouse monoclonal Ab (green), and rabbit polyclonal Abs (blue) were used as controls. Data shown here are a representative result of four independent experiments for each probe with similar results. **e** Comparison of β10-S and β10-S-β10 affinities in the SATiN system. The average $K_d$ values of four independent experiments for each probe are presented as mean ± SD. *P*-value was calculated using a two-tailed *t*-test. Source data are available in the Source Data file.

Fig. 1e) suggest a considerable difference of the probe binding affinities toward CR3022 antibody and general IgG. We therefore reasoned that the inhibition can be eliminated or alleviated simply by dilution, a step also required to reduce nonspecific signal when performing ELISA on blood samples. This was supported by computational simulation, using the obtained mathematical model, on virtual samples containing different concentrations of IgG and varying amounts of CR3022 (Supplementary Fig. 2a). The recovery rates (the luminescence ratio of a sample and the corresponding control without additional IgG) at different dilution endpoints were calculated (Supplementary Fig. 2b). At 1:100 dilution, the recovery rates for the samples of 20 mg/mL IgG (close to the upper limit in human serum) are around 20%. This was significantly improved by further dilution, reaching 47–60% at 1:300 dilution and 86–88% at 1:900 dilution. Reduced IgG brought even better recovery; samples containing 10 mg/mL IgG (close to the median IgG in human serum) showed ~80% recovery at 1:300 dilution and 5 mg/mL of IgG (close to the lower limit of IgG in human serum) showed more than 90% recovery at 1:300 dilution.

Building on these results, we then adopted a strategy involving sample dilution to obtain more accurate measurements of the S protein specific antibodies. The overall signal of a sample was calculated using the signal summation algorithm (luminescence sum of relative light unit (RLU) values at 1:300, 1:900, and 1:2700 dilutions) to avoid parameter estimation, as required by other algorithms, while still maintaining a power similar to curve fitting[32]. The signal at the lowest dilution endpoint (1:100) was excluded from the analysis to avoid the

substantial signal interference caused by IgG under these conditions. Computational simulation (Supplementary Fig. 2c) demonstrated good recovery; samples with medium and low IgG levels showed more than 80% recovery while samples with high IgG levels showed about 60% (for low CR3022 dose) to 70% (for high CR3022 dose) recovery. Performance was next evaluated experimentally with test samples containing varied doses of CR3022 and different concentrations of IgG (Fig. 2c); the results obtained were in good agreement with the computational prediction.

The performance was further evaluated via a spiking test with blank sera using seven serum samples collected before the pandemic. CR3022 was serially diluted and spiked into the matrix sera with concentrations of 100, 50, 25, 12.5, and 6.25 μg/mL. As before, SATiN assays were then performed on these samples by measuring the luminescence produced at further dilutions of 300, 900, and 2700 times, and the overall signal of each sample was calculated as a luminescence sum (Fig. 2d). The signals show an apparent linear relationship with the antibody concentration, indicating the detection range can reach to at least 100 μg/mL. Calculation using the criterion of 3 times the standard deviation of the blank samples suggests the limit of detection of the assay is below 5 μg/mL. Most of the recovery rates are within the range of 80–120% (Fig. 2d, inset), demonstrating the robustness of the assay. Since the samples contained serially diluted CR3022, the results also indicated a good dilution linearity. In addition, all inter-sample coefficients of variation of the groups with different amounts of spiked CR3022 were smaller than 20% demonstrating the good consistency of the assay.

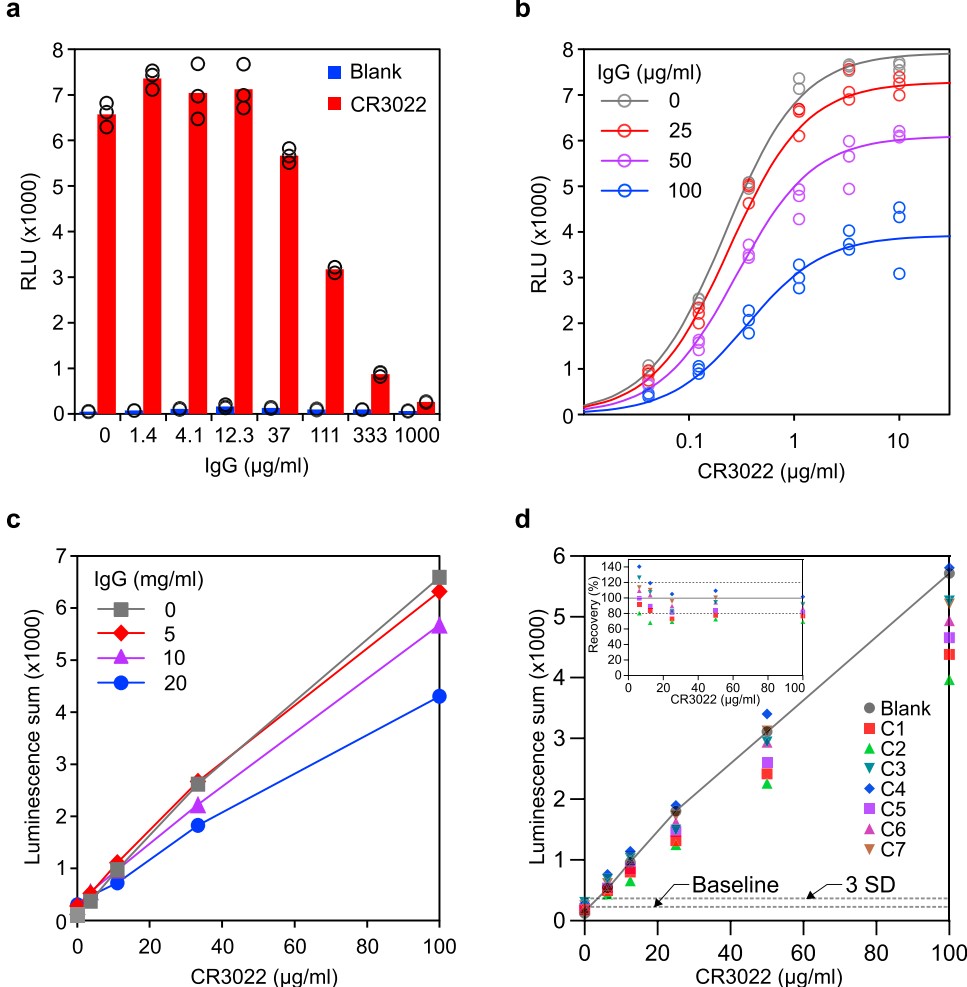

**Fig. 2 Characterization of SATiN assay. a** Inhibitory effect of additional IgG was examined using dose response. Different amounts of human IgG (as indicated) were applied to samples containing 2 μg/mL CR3022 (red) followed by analysis with the SATiN assay. Samples without CR3022 (blue) were used as negative control. Data shown here are a representative result of three independent experiments with similar results. **b** Inhibition kinetics of IgG in the assay were examined with different doses of CR3022, without IgG (gray) or in presence of human IgG at 25 (red), 50 (purple), or 100 (blue) μg/mL, roughly the amounts in human serum at a 1:200 dilution. Data shown here are a representative result of two independent experiments with similar results. **c** To mimic serum samples, different amounts of CR3022 (100, 33, 11, or 3.7 μg/mL) were spiked into buffer containing background human IgG at concentrations of 5 (red diamond), 10 (purple triangle), or 20 (blue circle) mg/mL, or without IgG (gray square). Each sample was serially diluted and then analyzed with the SATiN assay. Sums of luminescence readings at 1:300, 1:900, and 1:2700 for each sample are presented. Data shown here are a representative result of two independent experiments with similar results. **d** Serially diluted CR3022 (100, 50, 25, 12.5, and 6.25 μg/mL) was spiked into serum samples (n = 7, C1–C7 labeled with different colors and shapes) collected before the pandemic. In blank samples (gray circle), CR3022 was tested in buffer without mixing with serum. Each sample was tested and analyzed as in **c**. Basal value (baseline) was calculated as the mean of the seven serum samples without CR3022 spiking. The line of standard deviation (SD) × 3 is highlighted as the limit of detection. Recovery is calculated as the percentage signal of a sample divided by the corresponding blank sample and is plotted in the inset. Data shown here are a representative result of two independent experiments with similar results. Source data are available in the Source Data file.

**Evaluation of SATiN with human serum samples**. We then evaluated the assay with human serum samples. In addition to the above seven pre-pandemic sera, we also collected 82 serum samples from verified COVID-19 patients or convalescents across Canada, taken at different times (up to 80 days) after symptom onset. Again, the samples were serially diluted and analyzed using the SATiN assay. CR3022 was tested in parallel as a positive control. The samples displayed various signal amplitudes while those for pre-pandemic samples were close to baseline (Fig. 3a). As was done in Fig. 2c, d, the overall signal of the samples was calculated as luminescence summation of dilution points 1:300, 1:900, and 1:2700. A subset of the samples (n = 70) was re-tested with SATiN and comparison of the results obtained from these two independent tests showed a tight linear correlation

(Supplementary Fig. 3a), further demonstrating the consistency and reproducibility of the assay. Plotting the results against time of sample collection (Fig. 3b and Supplementary Fig. 3b) also provided a rough overview of the kinetics of IgG production/levels in the context of COVID-19 progression in patients; only background signals were detected in pre-pandemic samples, antibodies were detected as early as 4–5 days after symptom onset, all the samples collected during days 11–20 produced high level signals and the majority of samples collected after day 20 still contained high level of antibodies although a very small fraction dropped to basal levels. The results showed excellent separation between controls and patient samples and the kinetics were consistent with multiple previous reports[4,33–37], strongly suggesting good sensitivity and specificity of the SATiN assay.

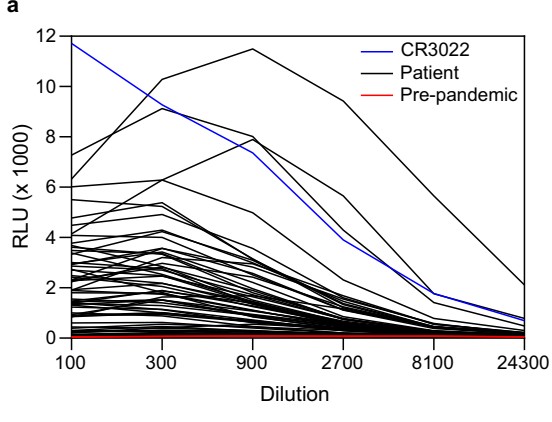
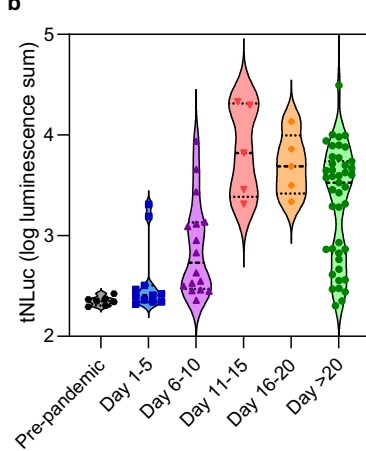

**Fig. 3 Detection of α-SARS CoV-2 antibody in serum samples with the SATiN assay. a** Samples included 7 (red) collected before the pandemic and 82 (black) from verified active or convalescent COVID-19 patients collected at different times after symptom onset. These were serially diluted as indicated and subjected to SATiN testing. CR3022 (0.4 mg/mL in stock, red) was used as positive control. The level of anti-SARS CoV-2 antibodies at each dilution measured using SATiN. Data shown here are a representative result of two independent experiments with similar results. **b**. The overall antibody signal in each sample was calculated by summation of luminescence signals at dilutions 1:300, 1:900, and 1:2700. Samples are categorized in groups based on time elapsed between symptom onset and sample collection, and their distribution is presented in violin plots. The central dashed lines represent medians. Source data are available in the Source Data file.

**Comparison of SATiN with other serological tests**. Finally, we compared the SATiN assay to several other serological tests. Similar to SATiN, ELISA directly measures the antibodies specific to SARS-CoV-2 in blood samples. We performed ELISA on 84 samples (82 verified COVID-19 patient or convalescent sera plus two pre-pandemic sera) using a common protocol[11]. The results demonstrated a high degree of correlation with those of the SATiN assay ($R = 0.886$) (Fig. 4a), suggesting that the quantifiability and sensitivity of SATiN are quite similar to ELISA. In contrast to direct ELISA, neutralizing antibody assays measure only the antibodies responsible for antagonizing virus–receptor interaction. We therefore carried out two neutralization assays on 80 of the samples: the surrogate virus neutralization test (sVNT)[38] (Fig. 4b) and the gold standard plaque-reduction neutralization test (PRNT50 and 90)[39,40] (Fig. 4c, d). The results from each of these tests showed high positive correlation with the SATiN assay, producing values of $R^2$ above 0.6 using Pearson analysis. These observations suggest that SATiN also provides a reliable indicator of the neutralization potential of a sample.

## Discussion

In this proof-of-principle study, we generated a serological assay called SATiN for the detection of antibodies against SARS CoV-2. Although follow-up studies on larger numbers of samples from different patients at various stages of disease progression are still required to fully characterize the SATiN system, the data in this work performed on sera derived from more than 80 COVID-19 patients have demonstrated the robustness of the assay. The underlying principle of SATiN allows the assay to be performed in homogeneous liquid phase, a feature granting rapid detection and exemption from many of the tedious procedures required by ELISA, but without compromising assay quantifiability or sensitivity. In addition, the SATiN assay can be scaled to high content measurement similar to many CLIAs. In contrast, however, it requires no specialized equipment beyond a regular microplate luminescence reader, making it highly cost-effective and suitable for both centralized and community testing. There is also the potential to modify SATiN for point-of-care testing.

Collectively these features suggest that our assay platform may be of great value for use in clinical laboratories involved in COVID-19 testing for diagnostics and surveillance. In addition, SATiN could be useful in other applications such as the clinical study of disease development and the evaluation of vaccination efficiency. In this work, we also characterized antibody kinetics using samples from a small cohort of patients, highlighting the value of SATiN as a tool for research. Another notable feature is that the SATiN assay readout shows a high degree of correlation with neutralizing capability, making it a potential platform to evaluate the antibody level induced by vaccination, although further assessment in this regard is still needed.

Although only IgG antibody testing has been described in this study, the principles of SATiN should also be applicable to the detection of antibodies of other immunoglobulin isotypes such as IgM and IgA. This might be performed by using their binders as sensor proteins, for example single chain antibodies, nanobodies, or similar molecules specifically recognizing these isotypes. Additionally, the SATiN strategy employed in our assay has the potential to be adapted for use in alternative diagnostic approaches to help detect responses to other pathogens and viruses such as SARS, MERS, and/or influenza.

## Methods

**Plasmids and recombinant proteins**. To create the general probe, the C2 domain of protein G was appended with the β9 tag and a 6xHis tag either at its N- or C-terminus (or both) (Supplementary Fig. 1a). The cDNAs (Supplementary Methods) were cloned into pET16b by Gibson assembly. The plasmids were transformed into BL21-Gold (DE3) cells. For expression, the bacteria were grown at 37 °C until OD550 0.4–0.6, followed by further incubation at 22 °C for 4 h in the presence of 0.2 mM IPTG. Bacterial cells were collected and lysed by sonication for further purification. Antibody-specific probes were created by appending β10 tag to N- or C-, or both termini of SARS-CoV-2 spike protein (ectodomain) or its RBD[41]. PDGFRB signal peptide was used as N-terminal leading sequence for N-tagged constructs to allow their correct secretion to extracellular space. PolyHis tag was added at their C-termini for purpose of purification. Several modifications were made to the S protein ectodomain probe similar to that previously reported: the foldon sequence was fused to the C-termini to promote trimerization; the furin cleavage site 682RRAR was mutated to GSAS; K986 and R987 were mutated to prolines to stabilize the prefusion conformation (Supplementary Fig. 1b). The plasmids of antibody probes were transfected to HEK 293 cells (a kind gift from Professor Jason Moffat, University of Toronto; originally purchased from the ATCC) by PEI and media of cultured cells were collected. Both general probes and

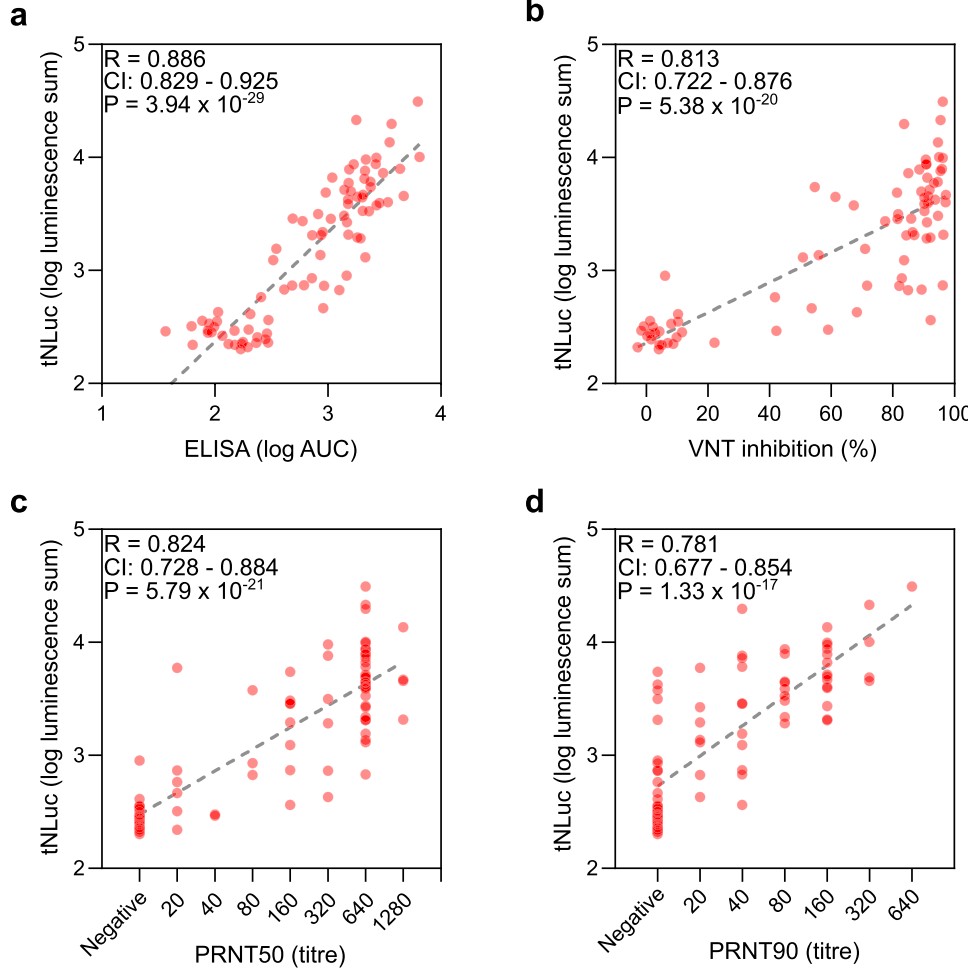

**Fig. 4 Comparison of SATiN assay with ELISA and neutralizing antibody assays. a** The 84 serum samples described in Fig. 3 were tested using ELISA and the results are compared with SATiN using a scatter plot. **b–d** Eighty sera were subjected to neutralizing antibody tests: sVNT (**b**), PRNT50 (**c**), and PRNT90 (**d**). *R* and *P*-values were obtained from two-tailed Pearson correlation analysis. Source data are available in the Source Data file.

antibody probe were purified with Ni Sepharose (Cytiva 17526801) according to the product manual. Δ11S was constructed as described[24] and purified as protein G probe as described above. The cDNA of CR3022 Fab was synthesized based on the reported sequence[42] and fused to human IgG1 Fc sequence to generate the IgG form. CR3022 was expressed in FreeStyle 293-F cells and purified from the culture medium using rProtein A Sepharose FF resin (GE Healthcare, 17127903). It was eluted from the resin with 50 mM glycine, pH 3.0, containing 150 mM NaCl followed by neutralization by the addition of 1/20 volume of 1 M Tris, pH 8.5.

**SATiN assay.** Patient sera or CR3022 were diluted in phosphate-buffered saline supplemented with 0.1% Tween-20 and 0.1% fatty acid-free bovine serum albumin. In some cases, human IgG (Sigma I4506) was added to mimic human serum sample. The samples were mixed with probes at a final concentration of 150 nM for general probes and 4 nM for antibody probes, and were incubated at room temperature for 30 min. An aliquot of the mixture (5 μL of serum sample or 2 μL of CR3022 sample) was mixed with 9 volumes of Nano-Glo buffer and substrate (Promega N1110) supplemented with recombinant Δ11S (final concentration 10 μg/mL) in 96- or 384-well microplates. After another 30 min of incubation at room temperature, luminescence signals were recorded by microplate luminometer Synergy LX (BioTek) and the data were collected using software Gen5 v3.09. CR3022 and probe binding kinetics were mathematically fit with GraphPad Prism 8 using an allosteric sigmoidal model. IgG inhibitory effects on CR3022 binding kinetics was fit using an allosteric non-competitive model:

$$Y = \frac{Y_{max}X^{n_x}}{X^{n_x}\left(1 + \frac{I^{n_i}}{\alpha K_i^{n_i}}\right) + K_m^{n_x}\left(1 + \frac{I^{n_i}}{K_i^{n_i}}\right)} \tag{1}$$

in which $Y$ is the luminescence signal, $X$ is the CR3022 concentration, $I$ is the IgG concentration, $n_x$ is the Hill coefficient for CR3022 and probe interaction, and $n_i$ is the Hill coefficient for probe and IgG interaction. The resultant global $R^2$ is 0.9911.

**ELISA.** ELISA was performed as previously described[10,11] with some modifications. SARS-CoV-2 spike antigen was immobilized to a 384-well LUMITRAC high binding plate (Greinor Bio-One 781074) by incubating 20 μL/well of S-β10 protein (20 nM in PBS) at 37 °C for 1 h. After blocking with a blocking buffer (PBS supplemented with 0.1% Tween 20 and 3% skimmed milk) for 1 h at room temperature, the plate was incubated with sera serially diluted in blocking buffer at room temperature for 2 h. After three times wash with PBST (PBS supplemented with 0.1% Tween 20), the plate was further incubated with α-human IgG antibody conjugated with horse radish peroxidase (Jackson ImmunoResearch 109-035-098, 1:50,000 in blocking buffer, 30 μL/well) at room temperature for 1 h. After three times final wash with PBST, Pico chemiluminescence substrate mixture (Thermo Scientific 3769, 30 μL/well) was added to the plate and the signals were recorded in a microplate luminometer. Data of serially diluted samples were analyzed with GraphPad Prism 9 by model fitting. The total signal was calculated as areas under curve (AUC) of the fitted curves.

**sVNT.** sVNT kit was purchased from Genscript and the assay was performed according to the manufacturer's instructions[38]. Briefly, diluted serum samples were mixed with HRP-RBD at a 1:1 ratio and incubated at 37 °C for 30 min. An aliquot (100 μL) of the mixture was moved to a well of a test plate which was precoated with ACE2 provided by the manufacturer, and was incubated at 37 °C for 15 min. After four times wash, thesignal was developed by incubation with TMB solution.

**PRNT.** PRNT was performed as described previously[43]. Briefly, diluted serum samples were incubated with SARS-CoV-2 virus (50 PFU) in a CO₂ incubator for 1 h. The mixture was then moved to a well of a 12-well plate cultured with Vero E6 cells (100% confluency) and incubated for 1 h in a CO₂ incubator with rocking every 15 min. To each well, 1.5 mL of prewarmed (37 °C) overlay medium (MEM without phenol red but supplemented with 4% FBS, ʟ-glutamine, nonessential amino acids, sodium bicarbonate, and 1.5% carboxymethycellose) was followed by incubation for 72 h. The cells were then fixed with 10% formalin (neutral-

buffered) and subsequently stained with crystal violet (0.5% solved in 20% ethanol). Plaques were counted and compared to negative control. A titer is recorded as the highest serum dilution resulting in 50% and 90% reduction in plaques compared with controls.

**Human serum samples.** Negative control sera were taken before the pandemic. All patients were diagnosed by SARS-CoV-2 RT-PCR from nasopharyngeal swabs. All samples are de-identified and enrolled through REB approved protocols, REB20-044c or REB 149-1994.

**Study approval.** The study was approved by the Office of Environmental Health & Safety at the University of Toronto. All research was performed in accordance with relevant guidelines and regulations. SATiN assay and ELISA were performed at the University of Toronto following the RIS protocol 40547 approved by the University of Toronto. sVNT and PRNT were performed at the National Microbiology Laboratory. External samples were transferred through Material Transfer Agreements. All participants have provided informed consent.

**Statistics.** Statistical analyses were performed using Microsoft Excel 2016 and GraphPad Prism 9. The performance difference between β10-S and β10-S-β10 was analyzed using an unpaired two-tailed $t$-test. Correlation between the SATiN assay and other assays (ELISA, VNT or PRNT) was analyzed using Pearson product–moment correlation.

**Reporting summary.** Further information on research design is available in the Nature Research Reporting Summary linked to this article.

## Data availability
All relevant data are available from the authors upon reasonable request. Source data are provided with this paper.

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

## Acknowledgements
We thank the members in Igor Stagljar's lab for their suggestions. This work was supported by the Toronto COVID-19 Action Fund (Connaught Award # 0000313897) to Igor Stagljar, and supported by intramural funding from the Office of the Vice President for Research and the 3i Initiative at the University of Utah to Shawn Owen.

## Author contributions

Z.Y. conceptualized the SATiN assay for antibodies against SARS-CoV-2, was actively involved in most experiments and data analysis, and wrote the bulk of the manuscript. L. D. and F.A. contributed to plasmid preparation and prepared the probes. Z.L. prepared the original cDNA for spike protein and produced CR3022. S.J.K. helped design the assay. H.W., E.J.V. and K.M. performed the neutralizing antibody assays. S.P., L.Y., X.L., Z.Z., F.Y.Y. and T.O. contributed to providing patient samples. J.S. contributed to manuscript writing and assay design. J.T. helped coordinate the project. M.A.D. coordinated the neutralizing antibody assays. A.M., M.O., S.J.D. and S.M. coordinated sample collection and processing. J.R. supervised the CR3022 production and contributed to manuscript writing. S.O. is the original inventor of tNLuc system and helped design the SATiN assay for antibodies against SARS-CoV-2. I.S. guided and supervised the work, contributed to assay development and manuscript writing as well as coordinated the preparation of the manuscript. All authors approved its content.

## Competing interests

A patent covering all of the main aspects/key elements of SATiN has been filed by the governing council of the University of Toronto (application number: US PROVISIONAL 63/121,689). The application is currently pending. I.S. and Z.Y. are named as co-inventors on the patent application. The remaining authors declare no competing interests.
