## [Peer Review File · Nature Communications]

Reviewers' Comments:

Reviewer #1:

Remarks to the Author:

In the manuscript Yao et al., have developed a method for detection of anti-SARS CoV-2 antibodies, based on a tri-part Nanoluciferase approach. The study is well described and provides a fast and sensitive approach for detection of anti-SARS CoV-2 antibodies, which is an urgent need for the society. The paper is based on the CR3022 antibody and sera from 12 convalescent COVID-19 patients (and 2 controls). The data show a good correlation to ELISA. The method described herein has the potential to be used for detection of other infectious agents and could, if developed to a commercial product, be of great use. The paper merits a rapid publication in its current state, due to the urgent need, although it would have been great to have a more thorough evaluation of the methods performance by increasing the number of patients and controls. Further, it would have been good with a discussion of approaches to also enable detection of other antibody isotypes, e.g. IgM.

Reviewer #2:

Remarks to the Author:

Yao and colleagues report on a novel tri-part Nanoluciferase immunoassay to detect IgG antibodies against the spike protein of SARS-CoV-2 in human serum. Their proof of principle study reports on a homogeneous immunoassay a tri-part NanoLuc® were diluted serum samples are mixed and incubated with two probes

(probe 1: B9 tag fused to the C2 domain of protein G; probe 2: B10 tag fused to the the SARS-CoV-2 spike protein) for 30 minutes in the first step and then an aliquote of this mixture is combined with the third component Δ11S and the luciferase substrate and incubated for another 30 min followed by

luminiscense recording with a luminator (tri-part Nano. The whole process occures directly in the liquid phase and does not require any waching steps or specialized equipment.

The findings of this manuscript are novel and of interest for the community. The manuscripts is very well written, methods are deccribed in detail, and the results and discussion give a good impression of the potential of the tri-part NanoLuc technology.

However, I have some comments:

1) The authors put great effort in developing this novel assay. However, clinical data is largely missing since 2 control sera and 10 sera from convalescent COVID-19 patients is not sufficient data to draw conclusion about similar sensitivities to ELISA or correlations with neutralizing antibody assays. Therefore, The authors should be carefull with the interpretation of their data that their "...proof-of-principle study suggests potential applications in diagnostics and disease and Vaccination management." (linies 47-48)

2) As stated by the authors several qualitative and quantitative SARS-CoV-2 andtibody CLIA assays have been approved, validated, and are widely available and routinely used in clinical practive. Please also change reference 12 (package insert) to on of the evaluation studies of the Roche Anti-SARs-CoV-2 assay. (lines 66-71)

3) I am somewhat confused aobut the approach to use different sample dilutions to obtain more accurate measurements of S protein specific antibodies (line 156 and following). When reporting results from conventional ELISA of CLIA I am used to standard dilutions and only if the antibody titer is to high further dilutions are performed. Please explain im more detail and or discuss in the discussion. Furthermore, I would assume the substantial signal interferences caused by IgG is the

reason for this signal summation algorithm and might be a drawback when compared to ELISA and or CLIA technology. Please clarify!

4) From an analytical point of view data on LOD, linearity, measurement range, CV's, etc. would be very interesting for such a novel assay.

5) From a clinical point of view more samples from healthy volunteers/blood donors (prepandemic) and specially more COVID-19 patients with different times from symptom onset and comparison to ELISA and neutralizing antibody assays would be necessary to get a better impression on the specificity and the sensitivity of the novel assay.

Point-by-point response

Response to Reviewer #1

Comment #1: *The paper merits a rapid publication in its current state, due to the urgent need, although it would have been great to have a more thorough evaluation of the methods performance by increasing the number of patients and controls. Further, it would have been good with a discussion of approaches to also enable detection of other antibody isotypes, e.g. IgM.*

Response to Comment 1: We would like to thank the reviewer for their comments. As suggested, we have now tested more samples reaching a total number of 89, including sera from 7 pre-pandemic individuals and 82 verified COVID-19 patients and convalescents (Fig. 3). We also appreciate the suggestion of using our system to detect other antibody isotypes, and have now commented in the Discussion on efforts currently being made in our lab in this regard.

Response to Reviewer #2

Comment #1: *The authors put great effort in developing this novel assay. However, clinical data is largely missing since 2 control sera and 10 sera from convalescent COVID-19 patients is not sufficient data to draw conclusion about similar sensitivities to ELISA or correlations with neutralizing antibody assays. Therefore, The authors should be carefull with the interpretation of their data that their "...proof-of-principle study suggests potential applications in diagnostics and disease and Vaccination management." (linies 47-48).*

Response to Comment 1: We fully agree with this comment. Unfortunately, due to the difficulty of obtaining clinical samples, we were only able to test 14 sera in our early work. This has been greatly improved in our revised manuscript, where we now have test data for 89 total samples (including sera from 7 pre-pandemic individuals and 82 COVID-19 patients and convalescents (Fig. 3A)). Analysis shows clear distinctions in tNLuc signals at different times of disease progression (Fig. 3B), suggesting good sensitivity and specificity of the assay. Comparison with ELISA and neutralizing assays using the new datasets displayed highly similar data trends, further confirming that the tNLuc assay (which we now call SATiN) has comparable sensitivity to ELISA and good correlation with neutralizing antibody assays.

Comment #2: *As stated by the authors several qualitative and quantitative SARS-CoV-2 antibody CLIA assays have been approved, validated, and are widely available and routinely used in clinical practice. Please also change reference 12 (package insert) to on of the evaluation studies of the Roche Anti-SARs-CoV-2 assay. (lines 66-71).*

Response to Comment 2: We appreciate the availability of many methods based on CLIA or similar principles and their strengths. As we described in the manuscript, however, these assays depend on highly specialized equipment and reagents and therefore are suitable for centralized testing. In contrast, the minimal requirement for our

SATiN assay is a luminometer, which allows it to be deployed in either a centralized testing format, or at a smaller and more economical scale for use in small communities/hospitals/labs etc. It can also potentially be modified for point-of-care applications. We therefore believe there are a lot of important use cases for our SATiN assay. Additionally, we have changed the reference as suggested by the reviewer.

Comment #3: *I am somewhat confused about the approach to use different sample dilutions to obtain more accurate measurements of S protein specific antibodies (line 156 and following). When reporting results from conventional ELISA or CLIA I am used to standard dilutions and only if the antibody titer is too high further dilutions are performed. Please explain in more detail and or discuss in the discussion. Furthermore, I would assume the substantial signal interferences caused by IgG is the reason for this signal summation algorithm and might be a drawback when compared to ELISA and or CLIA technology. Please clarify!*

Response to Comment 3: As suggested by the reviewer, endpoint titre is a routine method used in clinical practice. The drawback is that the resultant discrete numbers are usually not very powerful for detailed statistics analysis. In our current method development research, we sought to carefully evaluate our new assay and thereby decided not to take this approach.

A more accurate method is to fit data to a 4-parameter sigmoidal curve model and calculate the area under the curve (AUC), which is commonly used for research. This approach can be adapted to SATiN using a different mathematical model as described in the manuscript. Another prerequisite of the curve fitting approach is to know the IgG concentration in the sample (which actually can be measured using our SATiN assay). We did not take this approach, however, in order to avoid tedious experimental work and complicated computation.

Signal summation was first introduced as absorbance summation two years ago (Hartman et al. PLoS One. 2018;13(6): e0198528) and features simplicity and even better performance than curve fitting. In the short time since its publication, it has been used in multiple studies, including publications in Nature Immunology (22: 25) and JCI (130:3270). Using a signal summation approach makes our assay easier without compromising the study. It should be noted that the interference of IgG on SATiN is not the reason for the algorithm. We took the simple but effective step of dilution to relieve the interference. As we show in the manuscript, 300 times dilution has already significantly reduced the observed interference in most samples. Importantly, a similar problem also exists in ELISA in a different manner: high concentrations of IgG interfere with the real signal via nonspecific adsorption to the testing well. Thereby, dilution of up to 100-fold is usually required.

Although we use signal summation for high quality analyses in this study, we do not exclude endpoint titre. Our preliminary comparison with signal summation and endpoint titre indeed shows great correlation (please see the appended figure). We plan to collaborate with clinicians for further validation in clinical studies in the future. However, we feel that this exceeds the scope of

our current method development study and therefore do not address it in the manuscript.

Comment #4: *From an analytical point of view data on LOD, linearity, measurement range, CV's, etc. would be very interesting for such a novel assay.*

Response to Comment 4: We thank the reviewer for the suggestions. We used seven pre-pandemic sera as matrices to perform a spiking assay. The calculated LOD is lower than 5 µg/mL with a range up to 100 µg/mL. The assay demonstrated good linearity and excellent reproducibility as judged by low inter-sample CVs. Data are presented in the new **Fig. 2D** and described in the revised manuscript.

Comment #5: *From a clinical point of view more samples from healthy volunteers/blood donors (prepandemic) and specially more COVID-19 patients with different times from symptom onset and comparison to ELISA and neutralizing antibody assays would necessary to get a better impression on the specificity and the sensititiy of the novel assay.*

Response to Comment 5: We collected more pre-pandemic samples to reach a total number of seven. In addition, we added 70 more patient and convalescent samples in the revised study (**Fig. 3A**), including several samples collected within 5 days of symptom onset, most of which displayed basal level signals similar to pre-pandemic samples. The analysis of the SATiN signals against the times of sample collection (**Fig. 3B** and **Supplementary Fig. 3B**) is consistent with the kinetics described in many previous studies, further demonstrating the robustness of the assay. The comparison with ELISA and neutralizing antibody assay using the new samples (**Fig. 4**) further corroborated our conclusion.

Reviewers' Comments:

Reviewer #2:

Remarks to the Author:

My comments have been addressed adequately. I have no further comments.